# Patient-Derived Colorectal Cancer Organoids Upregulate Revival Stem Cell Marker Genes Following Chemotherapeutic Treatment

**DOI:** 10.3390/jcm9010128

**Published:** 2020-01-02

**Authors:** Rebekah M. Engel, Wing Hei Chan, David Nickless, Sara Hlavca, Elizabeth Richards, Genevieve Kerr, Karen Oliva, Paul J. McMurrick, Thierry Jardé, Helen E. Abud

**Affiliations:** 1Department of Anatomy and Developmental Biology, Monash University, Clayton Victoria 3800, Australia; rengel@cabrini.com.au (R.M.E.); horace.chan@monash.edu (W.H.C.); Sara.Hlavca1@monash.edu (S.H.); elizabeth.a.richards@monash.edu (E.R.); genevieve.kerr@monash.edu (G.K.); 2Stem Cells and Development Program, Monash Biomedicine Discovery Institute, Monash University, Clayton, Victoria 3800, Australia; 3Cabrini Monash University Department of Surgery, Cabrini Hospital, Malvern Victoria 3144, Australia; koliva@cabrini.com.au (K.O.); pjm@colorectal.com.au (P.J.M.); 4Anatomical Pathology Department, Cabrini Pathology, Cabrini Hospital, Malvern, Victoria 3144, Australia; David.Nickless@mps.com.au; 5Monash BDI Organoid Program, Monash Biomedicine Discovery Institute, Monash University, Wellington Road, Clayton, Victoria 3800, Australia; 6Centre for Cancer Research, Hudson Institute of Medical Research, Clayton, Victoria 3168, Australia

**Keywords:** bowel cancer, organoid, tumoroid, colorectal, colon, stem cell, chemotherapy resistance

## Abstract

Colorectal cancer stem cells have been proposed to drive disease progression, tumour recurrence and chemoresistance. However, studies ablating leucine rich repeat containing G protein-coupled receptor 5 (LGR5)-positive stem cells have shown that they are rapidly replenished in primary tumours. Following injury in normal tissue, LGR5+ stem cells are replaced by a newly defined, transient population of revival stem cells. We investigated whether markers of the revival stem cell population are present in colorectal tumours and how this signature relates to chemoresistance. We examined the expression of different stem cell markers in a cohort of patient-derived colorectal cancer organoids and correlated expression with sensitivity to 5-fluorouracil (5-FU) treatment. Our findings revealed that there was inter-tumour variability in the expression of stem cell markers. Clusterin (*CLU*), a marker of the revival stem cell population, was significantly enriched following 5-FU treatment and expression correlated with the level of drug resistance. Patient outcome data revealed that *CLU* expression is associated with both lower patient survival and an increase in disease recurrence. This suggests that *CLU* is a marker of drug resistance and may identify cells that drive colorectal cancer progression.

## 1. Introduction

Colorectal cancer (CRC) is the most frequently diagnosed cancer of the digestive tract and a principal cause of cancer-related deaths worldwide [1,2]. The majority of deaths from CRC can be attributed to cancer recurrence after initial treatment, which presents as distant metastases in secondary sites such as the liver or lung. If left untreated, the 5 year survival rate of patients with metastatic CRC can be as low as 5% [3,4]. Combined chemotherapy treatment, which includes the commonly used drug 5-fluorouracil (5-FU), can help to increase this survival rate [5,6]. However, development of more targeted and personalised treatments are necessary to decrease overall CRC mortality [7]. Understanding the mechanisms underlying resistance to common treatments incorporating 5-FU is complex and requires further elucidation in models that recapitulate the diversity of primary tumours arising [8].

Cellular heterogeneity is a feature of CRC tumours, where only a subset of cells display tumour-initiating activity [9,10]. The cancer stem cell (CSC) hypothesis supports a model where a small population of stem cells drive tumour growth and metastasis and may even predict disease relapse [11]. Furthermore, CSCs may enter a quiescent state, rendering them inherently resistant to anti-proliferative drugs. They may also stimulate tumour recurrence following therapy. [12]. CSC markers are promising prognostic biomarkers and therapeutic targets. However, several studies have now demonstrated the considerable plasticity of stem cell populations within tumours which creates further complexity when targeting these cell populations [13,14]. Characterising the dynamics of these cell populations in response to chemotherapeutic challenge is therefore of critical importance.

Normal intestinal crypt-based columnar (CBC) stem cell markers, including leucine rich repeat containing G protein-coupled receptor 5 (*LGR5*) and EPH receptor B2 (*EPHB2*), have been shown to be over-expressed in colorectal adenocarcinomas, and colorectal tumours that display a stem cell signature correlate with a decrease in disease-free survival of patients and an increase in relapse [15,16]. However, specific targeted ablation of LGR5+ cells in cancerous tissues revealed that the LGR5+ stem cell population is dispensable for primary tumour maintenance [13,14,17]. This is also the case when LGR5+ cells are deleted within the intestinal epithelium during normal homeostasis [17,18]. In normal tissues and tumours, it appears that cellular plasticity within intestinal cell populations allows LGR5- progenitors to revert back into a stem-like state and reconstitute the ablated LGR5+ stem cells [13,14,17,18]. In normal tissue, a quiescent reserve population of stem cells marked by BMI1 polycomb ring finger proto-oncogene (*BMI1*) has been postulated to be activated upon damage and regenerate lost *Lgr5+* cells [18,19]. Whether a population of cells with these characteristics are present in tumours is unclear. However, colon tumours with elevated levels of *BMI1* have been associated with reduced overall patient survival [20].

Recently, a novel unique stem cell population, marked by clusterin (*CLU*) and annexin A1 (*ANXA1*) expression, has been identified by single cell RNA sequencing in normal mouse small intestinal tissue. This stem cell pool is activated at the onset of tissue injury and repopulates the damaged small intestinal crypts, including the LGR5+ stem cell population [21]. The exact role of this revival stem cell population in a cancer context remains yet to be determined.

In this study, we aim to characterise the expression profile of the revival, CBC and quiescent stem cell markers in colorectal cancer and examine how this expression is modified upon chemotherapeutic treatment with 5-FU. To achieve this objective, we make use of patient-derived colorectal tumour organoids that were generated from treatment-naïve patients. This enables analysis of expression of stem cell markers in a cohort of organoid lines derived directly from patients. Subsequently, we investigate how the profile of different stem cell markers correlates with resistance to therapy and patient survival.

## 2. Experimental Section

### 2.1. Ethics and Consent

This study was conducted in accordance with the Declaration of Helsinki, and the protocol was approved by the Cabrini Human Research Ethics Committee (CHREC 04-19-01-15) and the Monash Human Research Ethics Committee (MHREC ID 2518). Patient recruitment was led by the colorectal surgeons in the Cabrini Monash University Department of Surgery. Tissue was obtained from treatment naïve patients diagnosed with colorectal cancer undergoing surgical resection at the Cabrini Hospital, Malvern, Australia. All subjects provided written informed consent.

### 2.2. Patient Data

Patient information including clinical characteristics, treatment regimen and outcome data (Appendix A) was obtained from the prospectively maintained, clinician-led Cabrini Monash University Department of Surgery colorectal neoplasia database (CMCND) [22]. This dataset has been adopted in a minimum dataset format as the Binational Colorectal Cancer Audit of the Colorectal Surgical Society of Australia and New Zealand (https://cssanz.org/bcca-database/).

### 2.3. Establishing Colorectal Cancer Organoids

CRC tissue specimens were cut into 5 mm pieces and washed eight times with 1× PBS supplemented with antibiotics. Tissue fragments were digested with 0.125 mg/mL dispase type II (Sigma, St Louis, MO, USA) and 1 mg/mL collagenase A (Roche Diagnostics, Mannheim, Germany) at 37 °C for 30 min and then mechanically dissociated by repetitive pipetting in cold PBS. Cancer tissue fragments were allowed to settle by gravity before supernatant was collected and pelleted by centrifugation at 240× *g* for 5 min at 4 °C. The isolated cells/fragments were passed through a 70 µm cell strainer (Corning, NY, USA), centrifuged and resuspended in Matrigel (Corning).

Matrigel containing cancer cell clusters were seeded into 24-well tissue culture plates (Thermo Scientific Nunc, Foster City, CA, USA) and allowed to polymerize for 10 min at 37 °C. The cancer cells were overlaid with 500 µL of culture medium composed of advanced Dulbecco’s modified Eagle medium/F12 supplemented with 1X B27, Glutamax, 10 mM HEPES (all from Gibco, Waltham, MA, USA), 100 µg/mL Primocin (InvivoGen, San Diego, CA, USA), 50 ng/mL recombinant human EGF (Peprotech, Rochy Hill, NJ, USA), 10 nM Gastrin (Sigma), 500 nM A83-01 (Tocris Bioscience, Bristol, UK), 1.25 mM N-acetylcysteine (Sigma), 10 mM nicotinamide (Sigma) and 100 ng/mL recombinant human Noggin (Peprotech) or 10% Noggin conditioned media, 20% R-spondin1 conditioned media. Following initial seeding of the cultures, 10 µM Y-27632 dihydrochloride kinase inhibitor (Tocris Bioscience) was also added to the media for 2–3 days.

### 2.4. Organoid Drug Sensitivity Testing

After the establishment of cancer-derived organoids, organoids were dissociated using TrypLE Express enzyme (Gibco) and re-seeded in Matrigel into a 48-well plate in triplicate. Organoids were cultured in complete media until small organoids were formed. Reference viability values were determined at day 0 by adding 200 µL of 1X Presto Blue reagent (Invitrogen, Carlsbad, CA, USA) diluted in culture medium to each well. Organoids were incubated for 45 min at 37 °C before the Presto Blue solution was removed into a black microplate and the fluorescence was measured (excitation of 560 nm and an emission of 590 nm) on the PHERAstar FS (BMG Labtech, Ortenberg, Germany). Complete media supplemented with 0, 0.1, 1, 10, 20 and 50 µm 5-fluorouracil (5-FU) (Sigma) was replaced onto the organoids at day 0 and day 2. Cell viability was measured at day 5, as for day 0.

### 2.5. Histological Sections

Primary tissue samples were fixed in 4% paraformaldehyde (PFA) and embedded in paraffin blocks. Mature organoids were fixed in 4% PFA before being dissociated from the Matrigel. Organoids were collected into a tube and gently centrifuged before being embedded into low melting agarose (2% diluted in PBS). The agarose blocks were processed before being embedded into paraffin. Sections (4 µm) of both primary tissue and patint-matched organoids were subjected to routine haematoxylin and eosin (H&E) staining.

### 2.6. Immunohistochemistry

The immunohistochemical procedure was conducted as previously described [23]. Briefly, slides were deparaffinized in histosol and rehydrated in graded alcohols. Antigen retrieval was performed by heating the slides for 10 min in a pressure cooker in 10 mM citrate buffer (pH = 6). Slides were blocked with CAS block (Invitrogen) for 1 h at room temperature. Sections were incubated overnight at 4 °C with the primary antibody diluted in PBS containing 1% bovine serum albumin. The following antibodies were used: anti-cleaved caspase 3 (Cell Signaling Technology, Danvers, MA, USA), anti-Cytokeratin 20 (Roche Ventana, Oro Valley, AZ, USA), anti-caudal type homeobox 2 (CDX2) (Abcam, Cambridge, UK) and anti-LGR5 ([24] gift from Dr Melissa R. Junttila, GenenTech, South San Francisco, CA, USA) (Table 1). For the detection of primary antibodies, sections were exposed to anti-goat or anti-rabbit horseradish peroxidase coupled antibodies (Life technologies, Carlsbad, CA, USA) in PBS with 1% bovine serum albumin for 1 h at room temperature. Peroxidase activity was detected with the 3, 3′-diaminobenzidine (DAB) liquid kit (Dako, Burlingame, CA, USA). Sections were counterstained with haematoxylin, dehydrated and mounted. Imaging was carried out using a Zeiss Axio Imager running ZEN digital imaging software (Carl Zeiss, Oberkochen, Germany).

### 2.7. Quantitative RT–PCR Analysis

Total RNA was isolated from tumour-derived organoids using the RNeasy Micro kit (Qiagen, Hilden, Germany). cDNA was synthesized from 400 ng of total RNA using QuantiTect Reverse Transcription Kit (Qiagen). Quantitative PCR was performed in technical triplicates on a LightCycler 480 II (Roche) using QuantiNova SYBR^®^ Green PCR Kit with thermal cycle as follows: 95 °C for 10 min, 45 cycles of 95 °C for 20 s, 60 °C for 30 s followed by 72 °C for 40 s. The average expression levels (2^−ΔCt^) for each gene was calculated relative to beta-2-microglobulin and β-actin expression levels. The primers used in the current study are listed in Table 2.

### 2.8. Survival Analysis

The survival analysis of The Cancer Genome Atlas (TCGA) data were performed using online tools OncoLnc (http://www.oncolnc.org/) [25] and SurvExpress (http://bioinformatica.mty.itesm.mx/SurvExpress) [26]. In OncoLnc, the correlation between *CLU* expression and prognosis of patient were analysed with the upper and lower 25 percentile of *CLU* expression in colon adenocarcinoma (COAD) patients (*n* = 220). Survival rate is represented by Kaplan−Meier plot and analysed using Log-rank test. In SurvExpress, Cox survival analysis with *CLU* expression was performed in Colon GSE41258 database (*n* = 244) and censored with both survival and recurrence data. A value of *p* < 0.05 was considered to be significant.

## 3. Results

### 3.1. Patient-Derived Colorectal Cancer Organoids Recapitulate the Histopathological Characteristics of Their Primary Tumours and Display Inter-Tumoural Heterogeneity in Stem Cell Signatures

In order to determine whether CRC patient-derived colorectal cancer organoids (PDCOs) are robust models for examining the expression pattern of stem cell markers, we first compared the biological characteristics of primary tumours and their matched PDCOs. Histological analysis of 10 primary tumours and their respective PDCOs (Figure 1A) was conducted by a trained pathologist. The histological profile of the primary tumour was generally well maintained in PDCOs, with organoids typically having a more cuboidal appearance, but present otherwise similar cellular morphology to the primary specimen. The primary tumour in patient 30T is a moderately differentiated colonic adenocarcinoma with the PDCO displaying a more cuboidal appearance with less nuclear pleomorphism evident. In tissue from 38T, the primary tumour is a well differentiated colonic adenocarcinoma and matched PDCOs appear more cuboidal and show less nuclear stratification, but are otherwise similar (Figure 1A). The primary tumour in 53T represents an intramucosal adenocarcinoma with luminal necrosis. Similar cell morphology and luminal necrosis is evident in the PDCO section (Figure 1A). In 63T, the primary tumour is a moderately differentiated adenocarcinoma displaying cuboidal cell morphology with PDCOs closely resembling this morphology (Figure 1A).

Commonly used for differential diagnosis of colorectal cancer, caudal type homeobox 2 (CDX2) is a transcription factor critical for intestinal development that is highly expressed in normal and neoplastic intestinal epithelium [27,28]. To confirm whether intestinal identity of epithelial tissue is maintained under culture conditions, we compared the expression of CDX2 in primary tumour and PDCOs. Strong and diffuse nuclear staining of CDX2+ was observed in all 10 paired primary tumour and PDCO sections, confirming the intestinal origin of these adenocarcinomas is maintained in culture. Representative images of CDX2+ staining are provided in Figure 1B.

The ability of PDCOs to recapitulate the profile of stem and differentiated cell populations of the primary tumour was examined by immunohistochemical analysis. The number of LGR5+ cells as well as the cytoplasmic staining intensity were variable across the primary tumour specimens. However, the levels of expression between the primary tumour and matched PDCO were comparable. Similarly, positive immunostaining for cytokeratin 20 (CK20), the protein encoded by the Keratin 20 (*KRT20*) gene, was observed in the cytoplasm and/or cell membrane of the PDCOs and the abundance of CK20+ cells closely resembled that of the patient-matched primary tumour. Representative images of LGR5 and CK20 staining are illustrated in Figure 1C.

To more broadly assess the expression profiles of stem cell markers in CRC, we performed qRT-PCR analysis on 10 PDCO lines, analysing basal expression levels of CBC stem cell markers *LGR5* and *EPHB2*, a quiescent stem cell marker *BMI1*, recently identified revival stem cell markers *CLU* and *ANXA1*, and a marker of differentiated intestinal cells, Keratin 20 (*KRT20*) (Figure 1D). These results were consistent with the immunostaining for LGR5 and CK20 (Figure 1C) and revealed unique expression profiles for each of the 10 PDCO lines. Given that it has previously been reported that aggressive CRCs are enriched in intestinal stem cell marker expression and this is predictive of relapse in CRC patients [16], we were interested in investigating whether the differential expression profiles of the PDCOs were predictive of chemotherapeutic drug response.

### 3.2. Elevated Expression of a Subset of Stem Cell Markers Correlates with Resistance to Chemotherapy

PDCOs from six treatment-naïve CRC patients with stage III (present in local lymph nodes) and IV (spread to distant organs) disease were treated with an increasing dose of common chemotherapeutic, 5-fluorouracil (5-FU). The 5-FU serves as the main backbone of adjuvant chemotherapy regimens for the treatment of CRC. Cell viability was assessed five days following initial treatment and was analysed by the slope of the dose-response curve (Figure 2A,B), and area under the curve (AUC) analysis (Figure 2C). The PDCOs display differential responses to 5-FU treatment including distinct phenotypic changes in organoids that are responsive to drug treatment (ORG54T) and those that are less sensitive to treatment (ORG64T) (Representative images; Figure 2A). The presence of cleaved caspase-3-positive apoptotic cells in 5FU-treated PDCO 54T further illustrates sensitivity to treatment compared with the absence of staining in treatment resistant PDCO 64T (Figure 2A, right panel).

To determine whether basal expression levels of stem cell markers in PDCOs are predictive of resistance to chemotherapeutic drug treatment, gene expression analysis (Figure 1D) was correlated with AUC (Figure 2C). We identified an inverse correlation with marker of intestinal differentiation, *KRT20* (*p* = 0.008). High *KRT20* expression was significantly associated with lower AUC values, conferring sensitivity to treatment. There is also a strong positive correlation with revival stem cell gene *CLU*, when *CLU* expression in PDCO 54T was eliminated as an outlier from analysis which fall away from the 95% prediction interval of the best-fit line. High expression of *CLU* was significantly correlated with high AUC values (*R*^2^ = 0.885; *p* = 0.004).

Following treatment with chemotherapeutic drug 5-FU, we analysed PDCO expression profiles for stem cell markers from the same six patient lines (Figure 2E). We performed qRT-PCR analysis of PDCOs five days following initial treatment with 0, 1 and 10 µM 5-FU. We observed a modest decrease in LGR5 expression at 1 µM (0.88-fold decrease; *p* = 0.0115). There was upregulation of expression at 10 µM for both *BMI1* (1.5-fold increase; *p* < 0.0001) and *KRT20* (2.4-fold increase; *p* < 0.0001). However, it was revival stem cell-associated genes *CLU* and *ANXA1* that were the most significantly upregulated, with expression robustly increasing up to 5-fold between the untreated (0 µM) and 10 µM treated PDCOs.

We further explored the prognostic relevance of *CLU* expression in CRC using the OncoLnc online system to analyse colon adenocarcinoma patient data from TCGA. Kaplan–Meier survival analysis revealed increased expression of *CLU* was associated with poorer survival outcomes (*p* = 0.0286) (Figure 3A). Using the SurvExpress online tool, we evaluated differences in overall survival and recurrence-free survival between the predicted high-risk and low-risk groups in the Sheffer series (GSE41258). The high-risk group that had decreased overall survival and increased disease recurrence was associated with significantly higher expression of *CLU* (*p* = 0.032 and 0.024 respectively) (Figure 3B,C), which is in line with previous observations [29,30].

## 4. Discussion

The initial objective of this study was to profile the expression of stem cell markers in a cohort of colorectal cancer-derived organoid cultures from different patients. We first determined how faithfully the organoids replicated the histological features of the primary tumour from which they were derived. The overall morphological features and level of differentiation was maintained in the PDCOs with some tumours displaying a moderate degree of differentiation and others with a more differentiated phenotype. This is consistent with previous studies conducted on PDCOs which suggest most epithelial features of the primary tumours are present in vitro [31,32,33,34]. All of the PDCOs were positive for the definitive intestinal marker CDX2 [28,35] and the expression of the CBC stem cell marker, LGR5 [36,37] and the cytokeratin marker CK20 was similar in the primary tissue and organoids. Some organoids displayed more widespread expression and this was reflected in the corresponding primary tissue, while others showed little expression of LGR5. This indicated that the PDCOs were closely modelling the epithelial features of the tumours from which they were derived. Subsequently, we examined individual PDCOs for the expression of stem cell markers. The results revealed considerable inter-tumour variability of expression of the different markers. Overall, this suggests marked heterogeneity between individual PDCOs. All of the cultures utilised in our study were derived from patient tumours naïve to treatment, so these initial results represent cellular expression without the selective pressure of chemotherapeutic treatment.

Using knowledge of the behaviour of different stem cell populations in normal tissue, we predicted that there may be a difference in the expression of the repertoire of markers following exposure to 5-FU. We treated each of the PDCO lines with 5-FU and measured cell viability to determine relative sensitivity and correlated this with the stem cell expression signatures. LGR5 clearly did not correlate with resistance to 5-FU. LGR5+ cells are extremely susceptible to damage induced by chemotherapy or radiation in normal mouse tissue and rapidly undergo apoptosis [17,18,19]. This is also observed in LGR5+ cells in primary intestinal tumours in mice and human colorectal tumours [13,14]. This suggests the susceptibility to treatment observed here and in other studies is a conserved feature of LGR5+ cells in both normal tissue and tumours. In contrast, markers of both quiescent reserve cells and the newly described revival stem cells were enriched upon exposure to 5-FU in PDCOs. In mouse models, BMI1+ cells appear to be more resistant to injury and lineage tracing experiments have revealed these cells are activated to replenish lost LGR5+ cells [18]. Revival stem cells cannot be detected in normal tissue and are only identified in a mouse injury model, where damage-induced revival stem cells are also capable of reconstituting LGR5+ stem cells and regenerating the intestinal epithelium [21]. Our study shows for the first time that a revival stem cell signature strongly correlates with chemoresistance in PDCOs and suggests that the process of re-population of LGR5+ cells observed in normal tissue, may also operate with a similar mechanism in human cancer. It remains to be elucidated whether a comparable hierarchical relationship between LGR5+ cells and CLU+ cells is conserved in human intestinal cancer tissues.

Patient-derived models of cancer, including PDCOs show great promise in trialling therapies before they reach the patient and evidence supporting how accurately these ex vivo models can predict patient response is beginning to emerge [32,38,39,40,41,42]. A recent report showed convincing data from ten patients that a viability assay could be used to predict patient response to irinotecan-based therapies using CRC-derived PDCOs [39]. However, the same testing protocol was not predictive for patient outcomes following treatment with 5-FU plus oxaliplatin. How predictive PDCO testing is for a range of therapies and what the most effective testing regime to implement is still being investigated. Here, we used PDCO testing to look at how markers of different stem cell populations behave following 5-FU treatment and identified *CLU*, a marker of the revival stem cell population, as being enriched. Interestingly, both the patients from whom PDCO cultures 64T and 67T were derived, which were most resistant in our assay of sensitivity to 5-FU, have progressive disease. These PDCOs also exhibited the highest levels of *CLU* expression following treatment. Although the sample size is limited, this provides the basis for future studies on the role of revival stem cells in progressive disease in colorectal cancer patients. This cell type may be a potential therapeutic target and a marker of drug resistance.

## 5. Conclusions

Our study demonstrates that PDCOs are relevant in vitro models for studying the heterogeneity of stem cell populations represented in the primary tumours. We show that there is considerable variation between individual PDCOs in the repertoire of stem cell markers present. We have identified that *CLU*, a marker of the revival stem cell population, is enriched in PDCOs resistant to 5-FU and this is consistent with overall patient data showing that *CLU* correlates with lower survival and an increase in disease recurrence.

## Figures and Tables

**Figure 1 jcm-09-00128-f001:**
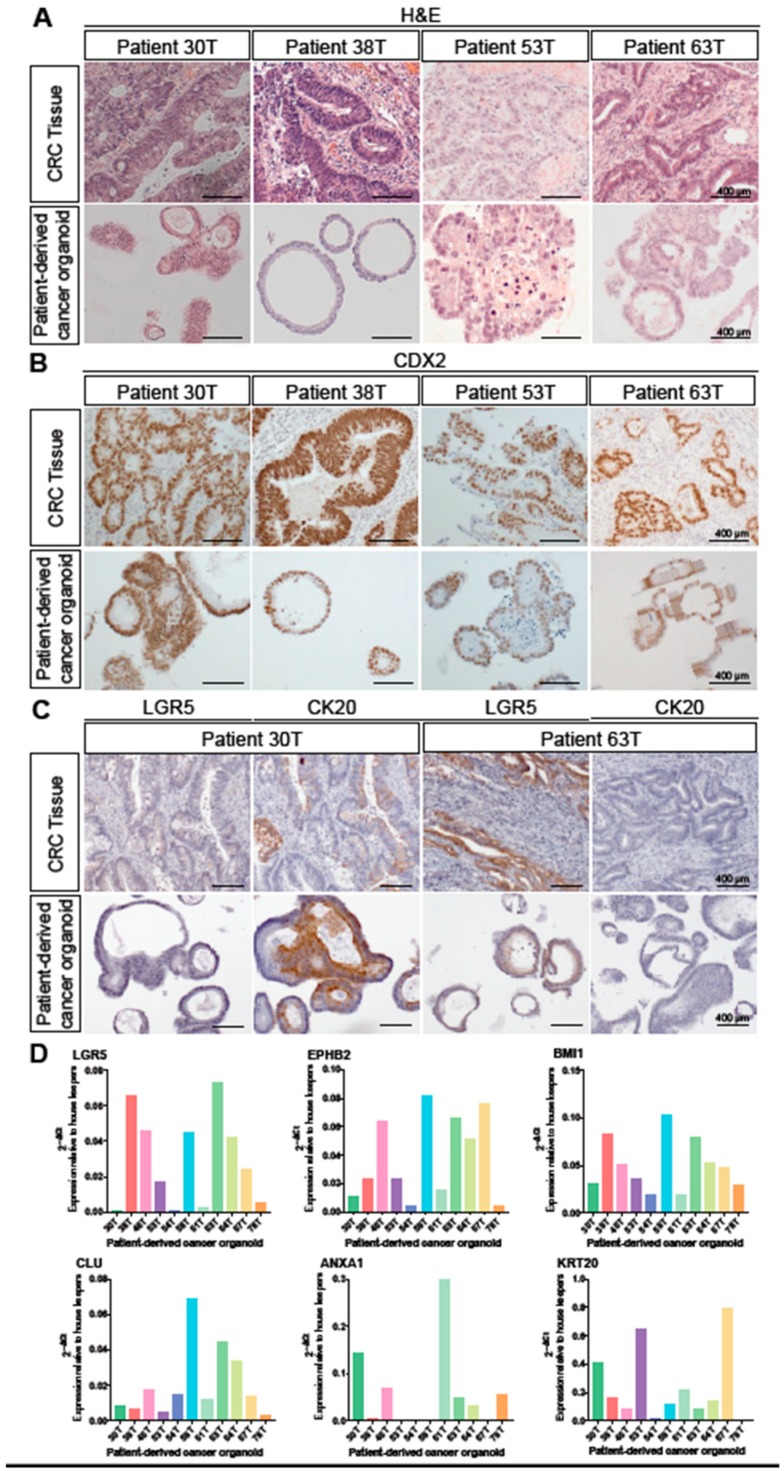
Patient-derived cancer organoids recapitulate the histopathological characteristics of their primary tumours and inter-tumoural heterogeneity in stem cell signatures. (**A**): Haematoxylin and eosin (H&E) staining of sectioned tissue from primary colorectal adenocarcinoma and patient-derived cancer organoids derived from the same tumour. Scale bar, 400 μm. (**B**): Immunohistochemical detection of CDX2 (marker of adenocarcinomas of intestinal origin) in primary colorectal adenocarcinoma compared to patient-derived colorectal cancer organoids (PDCOs). Scale bar, 400 μm. (**C**): Immunohistochemical detection of LGR5 (CBC stem cell marker) and CK20 (intestinal epithelial marker) in the colorectal adenocarcinomas and PDCOs. Scale bar, 400 μm. (**D**): The expression levels (2^−ΔCt^) for *LGR5, EPHB2, BMI1, CLU, ANXA1* and *KRT20* were calculated relative to beta-2-microglobulin and β-actin by qRT-PCR in individual PDCO lines.

**Figure 2 jcm-09-00128-f002:**
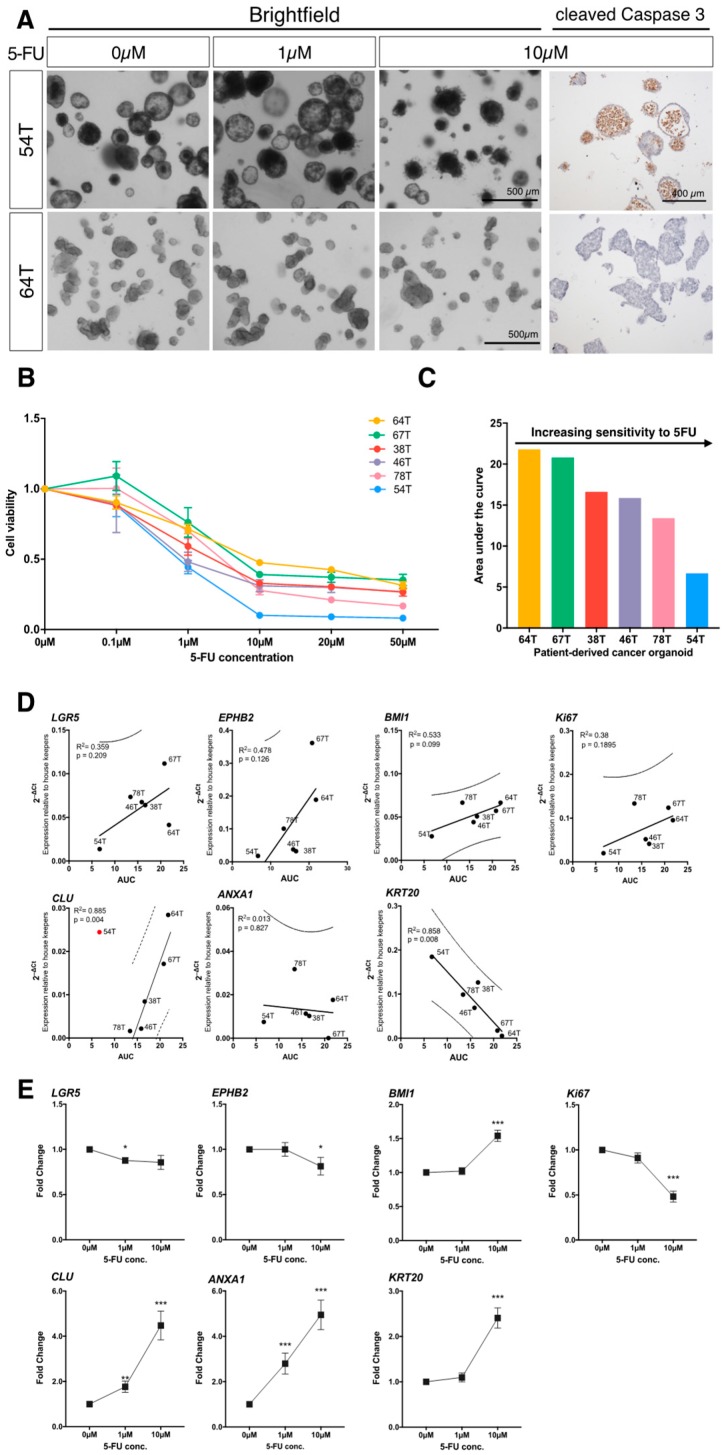
Elevated expression of a subset of stem cell markers correlates with resistance to chemotherapy. (**A**): Representative brightfield images of patient-derived cancer organoids (PDCOs) in response to chemotherapeutic treatment with 5-fluorouracil (5-FU) and the immunohistochemical detection of the active cleaved form of caspase 3 (apoptotic marker) in 10 µM treated PDCOs. Scale bar, 500 μm. (**B**): Dose-response curve of PDCOs in response to chemotherapeutic treatment with 5-FU (*n* = 6, mean ± SEM). (**C**): Area under the curve (AUC) analysis of 5-FU sensitivity in six patient-derived tumoroids. (**D**): Linear regression correlation analysis between the expression of stem cell marker genes and AUC showing the best-fit line (solid line) and 95% prediction interval (dash-line). (**E**): Expression of stem cell marker genes in PDCOs in response to an increasing dose of 5-FU. Fold change is calculated by the average gene expression levels (2^−ΔCt^) relative to the vehicle control (*n* = 6, mean ± SEM). In each graph, asterisks indicate pairs of means (compared to vehicle control) that were significantly different using Mann–Whitney test (*, *p* < 0.05; **, *p* < 0.01; ***, *p* < 0.001).

**Figure 3 jcm-09-00128-f003:**
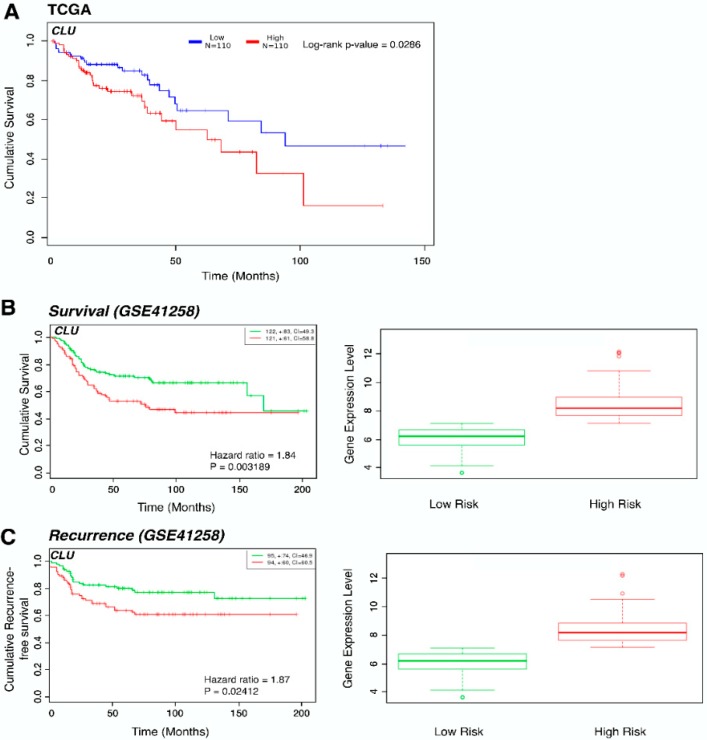
*CLU* expression in colon adenocarcinoma patients is associated with decreased overall survival. (**A**): Kaplan–Meier survival plot comparing The Cancer Genome Atlas (TCGA) colon adenocarcinoma patients with high (*n* = 110) and low (*n* = 110) expression of *CLU* using the OncoLnc tool. The associated log-rank *p*-value is 0.0286. (**B**): Kaplan–Meier survival plot for high- (red, *n* = 121) and low- (green, *n* = 122) risk groups in GSE41258 database by SurvExpress tool shows cumulative survival against time (months) and the box plot shows the corresponding *CLU* expression across groups. The number of individuals, the number of censored, and the CI of each risk group are shown in the top-right insets. Censoring samples are shown as “+” marks. (**C**): Kaplan–Meier survival plot for high- (red, *n* = 95) and low- (green, *n* = 94) risk groups in GSE41258 database by SurvExpress tool shows cumulative recurrence-free survival against time (months) and the box plot shows the corresponding *CLU* expression across groups. The number of individuals, the number of censored, and the CI of each risk group are shown in the top-right insets. Censoring samples are shown as “+” marks.

**Table 1 jcm-09-00128-t001:** List of antibodies used for immunohistochemistry.

Antibodies/Dye	Host	Dilution	Supplier	Cat/Lot number
**Primary**				
Cleaved Caspase 3 (Asp175)	Rabbit	1:250	Cell Signaling	9661S
Cytokeratin 20 (CK20) (SP33)	Rabbit	1:1	Roche Ventana	790-4431
Caudal type homeobox 2 (CDX2)	Rabbit	1:1000	Abcam	Ab76541
Leucine rich repeat containing G protein-coupled receptor 5 (LGR5)	Rabbit	1:200	Genentech	n/a
**Secondary**				
Anti-rabbit horseradish peroxidase	Goat	1:200	Life Technologies	G21234

**Table 2 jcm-09-00128-t002:** List of primers used for qPCR analysis.

Gene	Forward Primer Sequences	Reverse Primer Sequences	Product Length (bp)
*ACTB*	CTGGCACCACACCTTCTACAATG	GGTCTCAAACATGATCTGGGTC	124
*ANXA1*	TTTGCAAGAAGGTAGAGATAAAGAC	GGATGACTTCACAGTTTGAACAT	121
*B2M*	GTGCTCGCGCTACTCTCTC	GTCAACTTCAATGTCGGAT	142
*BMI1*	GGTACTTCATTGATGCCACAACC	CTGGTCTTGTGAACTTGGACATC	104
*CLU*	CAGGCCATGGACATCCACTT	GTCATCGTCGCCTTCTCGTA	78
*EPHB2*	TTGGGCTCTCACGCTTTCTA	AGGTGAACTTCCGGTACTGG	120
*Ki67*	CAGCACCTGCTTGTTTGGAAG	TAATATTGCCTCCTGCTCATGGAT	109
*KRT20*	CTGAGGTTCAACTAACGGAGCTG	AACAGCGACTGGAGGTTGGCTA	129
*LGR5*	CCTTCCAACCTCAGCGTCTT	AGGGATTGAAGGCTTCGCAA	250

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
