# Peer review of "Patient-Derived Colorectal Cancer Organoids Upregulate Revival Stem Cell Marker Genes following Chemotherapeutic Treatment"

_jcm, 2020, doi:10.3390/jcm9010128_

Round 1
Reviewer 1 Report
In this manuscript titled “Patient-derived colorectal cancer organoids 2 upregulate revival stem cell marker genes following 3 chemotherapeutic treatment”; Engel et al analyzed the colon cancer stem cells in general and studied the expression of stem cell marker CLU, a marker of the revival stem cell population. They used patient derived colorectal cancer organoids and analyzed the expression of different stem cell marker. Being responsible for their resistance to conventional chemotherapeutics CSCs are an important subpopulation to study. Therefore, the question that is being addressed in this manuscript is highly important and interesting however, there are some flaws in the terms of providing only a limited experimental evidence to support their conclusion. In general authors have tried to address a very relevant question, however more information are needed to support their conclusion. Considering the immediate need of increasing our understanding about role of CSC population, it is highly interesting to assess the expression and individual contribution of various cancer stem cells marker. Altogether, their preliminary result are interesting however, authors need more evidence to establish their conclusion.
My Specific comments are as follows:
In the figure 1, authors conclude that patient derived cancer organoids recapitulate the histo-pathological of their primary tumors and inter tumor heterogeneity. Authors need to characterize these organoids in terms of expression of conventional colon CSC markers like LGR5 and CD133 to establish the real heterogeneity and compare it with their primary tumors. In figure 2 they need to show the elevated expression of CSC markers at endogenous level by Western blot and by flow cytometery for surface markers in particular. Authors conclude that CLU is enriched in 5-FU resistant PDCOs. They need to explain if these CLU enriched population are derived from LGR5 positive population or not with proper experimental proof.
Reviewer 2 Report
Authors demonstrated Relationship between 5FU resistance and increase of a new revival stem cell marker CLU.
Their findings underly the stemness heterogeneity of colon tumor and in consequence tumor primary cell culture or tumor organoid to screen new drug and/or to develop personalized médicine.
Specific comment to authors:
Figure 2A: replace Caspase -3 by cleaved Caspase 3
Figure 2C : add an Arrow from left to right that indicated "increase of sensitivity to 5FU " or an Arrow from the right to the left indicated "incease of resistance to 5FU "
line247: Authors should indicate AUC (Air Under the Curve)
Line 249 and Figure 2D: Authors should explain the "aberrant" value of 54T because a part of their demonstration is on this sample.
Line 251 and Figure 2E: Authors should describe what are the sample considered and how many tumors are used.
Figure 2E: **** doesn't exist in international nomenclature for now even if "graphpad" or other software write it only *, or ** or *** exist. Please correct this mistake.
In the discussion Authors should indicate also this reference (PMID: 26791256)
Author Response
Response to Reviewer 2 comments
Point #1: Figure 2A: replace Caspase -3 by cleaved Caspase 3
Response #1: We have revised figure 2A and replaced Caspase -3 by cleaved Caspase -3.
Point #2: Figure 2C: add an Arrow from left to right that indicated "increase of sensitivity to 5FU" or an Arrow from the right to the left indicated "increase of resistance to 5FU"
Response #2: We have revised the figure and added an arrow pointing from left to right to indicate the increasing of sensitivity to 5FU.
Point #3: line247: Authors should indicate AUC (Air Under the Curve)
Response #3: We have now mentioned “Area Under the Curve (AUC)” in line 238.
Point #4: Line 249 and Figure 2D: Authors should explain the "aberrant" value of 54T because a part of their demonstration is on this sample.
Response #4: The CLU expression data of 54T fall outside of the 95% prediction interval of the best-fit line when eliminated from the regression plot. Therefore, we considered it as an outlier for reporting the correlation between CLU expression and Area under the curve. We have revised the main text to explain this and revised figure 2D to include the 95% prediction interval.
Point #5: Line 251 and Figure 2E: Authors should describe what are the sample considered and how many tumors are used.
Response #5: We have amended the text to indicate that this analysis is performed on PDCOs “from the same six patient lines”. In the text above, we have described these PDCOs; “…from six treatment-naïve CRC patients with stage III (present in local lymph nodes) and IV (spread to distant organs) disease”. The figure legend states that n=6 was used for this experiment.
Point #6: Figure 2E: **** doesn't exist in international nomenclature for now even if "graphpad" or other software write it only *, or ** or *** exist. Please correct this mistake.
Response #6: We have now removed **** from the figure legend.
Point #7: In the discussion Authors should indicate also this reference (PMID: 26791256).
Response #7: This reference is now included in the discussion after the following statement: “Patient-derived models of cancer, including PDCOs show great promise in trialing therapies before they reach the patient and evidence supporting how accurately these ex vivo models can predict patient response is beginning to emerge.”
Round 2
Reviewer 1 Report
The authors have addressed my concerns within the limit of the scope of the current manuscript and their available resources.